# In-Season Internal Load and Wellness Variations in Professional Women Soccer Players: Comparisons between Playing Positions and Status

**DOI:** 10.3390/ijerph182312817

**Published:** 2021-12-05

**Authors:** Renato Fernandes, João Paulo Brito, Luiz H. Palucci Vieira, Alexandre Duarte Martins, Filipe Manuel Clemente, Hadi Nobari, Victor Machado Reis, Rafael Oliveira

**Affiliations:** 1Sports Science School of Rio Maior–Polytechnic Institute of Santarém, 2040-413 Rio Maior, Portugal; jbrito@esdrm.ipsantarem.pt (J.P.B.); alexandremartins@esdrm.ipsantarem.pt (A.D.M.); 2Life Quality Research Centre, 2040-413 Rio Maior, Portugal; 3University of Trás-os-Montes e Alto Douro, 5001-801 Vila Real, Portugal; victormachadoreis@gmail.com; 4Research Centre in Sport Sciences, Health Sciences and Human Development, 5001-801 Vila Real, Portugal; 5Graduate Program in Movement Sciences, MOVI-LAB Human Movement Research Laboratory, Physical Education Department, School of Sciences, UNESP São Paulo State University, Bauru 17033-360, Brazil; luiz.palucci@unesp.br; 6Comprehensive Health Research Centre (CHRC), Departamento de Desporto e Saúde, Escola de Saúde e Desenvolvimento Humano, Universidade de Évora, Largo dos Colegiais, 7004-516 Évora, Portugal; 7Escola Superior Desporto e Lazer, Instituto Politécnico de Viana do Castelo, Rua Escola Industrial e Comercial de Nun’Álvares, 4900-347 Viana do Castelo, Portugal; filipe.clemente5@gmail.com; 8Instituto de Telecomunicações, Delegação da Covilhã, 1049-001 Lisboa, Portugal; 9Department of Physical Education and Sports, University of Granada, 18010 Granada, Spain; hadi.nobari1@gmail.com; 10HEME Research Group, Faculty of Sport Sciences, University of Extremadura, 10003 Cáceres, Spain; 11Department of Exercise Physiology, Faculty of Educational Sciences and Psychology, University of Mohaghegh Ardabili, Ardabil 56199-11367, Iran; 12Sports Scientist, Sepahan Football Club, Isfahan 81887-78473, Iran

**Keywords:** muscle soreness, female, stress, fatigue, sleep, perceived exertion, training monotony, training strain

## Abstract

The internal intensity monitoring in soccer has been used more in recent years in men’s football; however, in women’s soccer, the existing literature is still scarce. The aims of this study were threefold: (a) to describe the weekly variations of training monotony, training strain and acute: chronic workload ratio through session Rated Perceived Exertion (s-RPE); (b) to describe weekly variations of Hooper Index [stress, fatigue, Delayed Onset Muscle Soreness (DOMS) and sleep]; and (c) to compare those variations between playing positions and player status. Nineteen players (24.1 ± 2.7 years) from a Portuguese BPI League professional team participated in this study. All variables were collected in a 10-week in-season period with three training sessions and one match per week during the 2019/20 season. Considering the overall team, the results showed that there were some associations between Hooper Index categories and s-RPE like stress or fatigue (0.693, *p* < 0.01), stress or DOMS (0.593, *p* < 0.01), stress or s-RPE (−0.516, *p* < 0.05) and fatigue or DOMS (0.688, *p* < 0.01). There were no differences between all parameters in playing positions or player status. In conclusion, the study revealed that higher levels of fatigue and DOMS occur concurrently with better nights of sleep. Moreover, any in-season variations concerning internal load and perceived wellness seems independent of position or status in outfield players. The data also showed that the higher the players’ reported stress, the lower the observed s-RPE, thus possibly indicating a mutual interference of experienced stress levels on the assimilation of training intensity by elite women soccer players.

## 1. Introduction

Load/intensity monitoring is a well-implemented practice in team sports that guides the coach’s interventions through a better understanding of the impact of training stimulus on players [1,2]. Monitoring consists of using a given instrument or technique that allows us to track the intensity of exercise in each player. Thus, a wide vision of how the player is coping with the training process is implementing a strategy in which training intensity monitoring is complemented by wellness and readiness monitoring [3]. This integrated approach, also known as athlete’s monitoring allows understanding of the mechanisms related to training stimulus and recovery, thus providing some information about how the training periodization is actually done while helping coaches to quickly identify the individual responses of players to stressful situations while monitoring their wellness [4].

Due to the evident variations of a competitive schedule and as a part of the training methodology, it is expected to have in seasonal variations of training intensity (relative or absolute) in players [5]. Training demands are closely related to the training goals and structure of exercise; namely organization, quality, and quantity [2]. Thus, training intensity can be understood as an input variable constrained by the organization of training activities to elicit a given training response in accordance with the coach’s expectations [2]. Training intensity monitoring can be organized in two main dimensions: external and internal. The contextualization of these two dimensions is important, since internal responses (i.e., psychophysiological responses) are closely related with the physical demands imposed by a given training drill.

The external demands represent the immediate player’s physical responses to the organization, quality, and quantity of exercise (training plan) [2]. In the case of team sports, such physical responses are commonly analyzed using microelectromechanical systems or optical systems that provide estimated values related to distance-based, accelerometry-based, and combined variables [1]. These devices have been used not only in soccer, but in different women’s team sports (e.g., rugby, volleyball, handball) in which distance-based and accelerometry-based measures are important to understand the dynamics of training in an heterogenous groups [6,7]. Currently, the use of microelectromechanical systems (e.g., global positioning systems, local positioning systems or inertial measurement units) allows us to individualize the understanding of intensity demands in women sports such as rugby [8] or soccer [9]. While the external intensity is the acute physical response to the session training plan, the internal demands can be understood as the psychophysiological response to the external demands [10]. This means that, although there are possible similarities in external load, the internal load can be considerably different between two players. Internal responses can be constrained by the individual characteristics of the players, training status, psychological status, health, nutrition, environment, or genetics [2]. Thus, it is reasonable to predict that the same training plan, with possible similar external loads, can provide different internal loads in players. While internal demands represent the acute psychophysiological responses to the exercise, it is expectable that the consistency of the internal training demands across time leads to adaptations in the physiological levels of players with natural variations in the training outcomes [11,12,13].

Internal intensity is a crucial part of the training monitoring processes. Among different possibilities for monitoring the internal measures (e.g., heart rate monitors, respiratory gas analyzer, and blood lactate), the Rated Perceived Exertion (RPE) is the easiest instrument to apply, since it ensures a valid, reliable, and sensitive approach to quantify and qualify the internal load while using a simple questionnaire [14,15]. Moreover, the final score obtained by the RPE questionnaire can provide useful information to estimate the internal intensity (namely, multiplying the RPE score by the time of training in minutes) [16], or even using this RPE-based training load to estimate the organization of the training, calculating the variability of the load applied in the week (e.g., using an equation to estimate the Training Monotony [TM]) [17], the progression of load across the weeks (e.g., using the Acute: Chronic Workload Ratio [ACWR] to identify a measure of increase of decrease of a load of a week in comparison to the previous one) [18] or identify the Training Strain [TS] of loads during a week (e.g., multiplying the training monotony by the acute load of the week) [17].

Although some research claims cause–effect consequences of specific intensity measures and injury occurrence [19], this has been dismissed since the absence of quality data and proper methodological approaches to prove that [20,21,22]. However, these intensity measures are still valuable for guiding coaches to understand the dynamics of stimulus and the impact of the training plans on the actual responses of the players as recently shown in women soccer players [23]. The main evidence about internal intensity variations across the season, while using RPE-based measures, suggests that pre-season is the period in which the internal intensity accumulated in the week and the intensity measures of monotony and strain are typically greater than in the in-season periods [24]. This can be related with the higher external demands occurring in such a period [25], as a consequence of the typical strategy of increasing the volume and frequency of training sessions to provide a higher stress on the player’s organism before starting competition.

Although no strong relationships between training intensity and wellness are identified [26], wellness can be related to different variables, thus we can assist with variations across the season. In the context of soccer monitoring, wellness is quantified and qualified by the Delayed Onset Muscle Soreness (DOMS), sleep quality, fatigue, stress and mood, partially justifying these outcomes by the proposal of Hooper [27] which were updated by MacLean in 2010 [28]. Wellness is currently analyzed using questionnaires and those seem to be sensitive to variations within and between weeks (or periods). As an example, comparisons between pre-season and in-season revealed that pre-season was more strenuous and exhausting for players than the in-season period regarding the wellness variables that were inspected [29].

The main evidence on training monitoring and wellness in soccer has been related to men. However, women soccer players and training monitoring is still growing and due to the natural biological differences, more research to understand the mechanisms of how they cope with training process is needed. Thus, descriptive studies, namely cohorts, are still valuable for characterizing the reality of the training process in women’s soccer. Based on that reason, the purposes of this study were threefold: (a) to describe the weekly variations of TM, TS and ACWR through s-RPE; (b) to describe weekly variations of stress, fatigue, DOMS and sleep; and (c) to compare those variations between playing positions and player status.

## 2. Materials and Methods

### 2.1. Subjects

Considering several studies conducted with small sample sizes [5,11,12,13,23,25,29,30,31,32,33,34,35,36], 19 professional female soccer players participated in this study (24.1 ± 2.7 years, 164.3 ± 4.2 cm, 58.5 ± 8.2 kg). The players belong to a team that participated in the Portuguese BPI League in the 2019/20 season. Based on player status, they were divided into two groups: starters (*n* = 11) and non-starters (*n* = 8). Additionally, the playing positions were divided into five defenders, five central midfielders, four wide midfielders and five strikers. The inclusion criteria included regular participation in most of the training sessions (80% of weekly training sessions), while the exclusion criteria included lack of player information, illness and/or injury for two consecutive weeks. Goalkeepers were excluded from the study. The criteria to define starters and non-starters were assessed week by week against a player’s attendance time at the match and training sessions, and to be considered a starter, a player had to complete at least 60 min in three consecutive matches; players who did not achieve this duration were considered non-starters [34]. All participants were familiarized with the training protocols and signed informed consent prior to the investigation. This study was conducted according to the requirements of the Declaration of Helsinki and was approved by the Ethics Committee of Polytechnic Institute of Santarém (252020 Desporto).

### 2.2. Design

Training and match data were collected over a 10-week in-season period (between October and December) with three training sessions and one match per week. For the purposes of the present study, all the sessions carried out as the main team sessions were considered. This refers to training sessions in which both the starting and non-starting players trained together. Data from rehabilitation or additional training sessions of recuperation were excluded. This means that sessions after the match day were included whenever both starters and non-starters trained together, but other kinds of recovery training were excluded. This study did not influence or alter the training sessions in any way. Training data collection for this study was carried out at the soccer club’s outdoor training pitches. Accumulated total minutes of all training sessions per week are presented in Table 1. Each training session included the warm-up, main phase and slow-down phase plus stretching.

### 2.3. Internal Training Load/Intensity Quantification

During training sessions, the CR10-point scale, adapted by Foster et al., was applied [16]. Specifically, thirty minutes after the end of each training session, players rated their RPE value using an app on a tablet. The scores provided by the players were then multiplied by the training duration to obtain the s-RPE [16,37]. The players were previously familiarized with the scale, and all answers were provided individually to avoid non-valid scores.

### 2.4. Wellness Quantification

Approximately 30 min before each training session, each player was asked to provide the Hooper Index (HI) scores using an app on a tablet. This index includes four categories: fatigue, stress, muscle soreness (scale of 1–7, in which 1 is very, very low and 7 is very, very high), and the quality of sleep of the night that preceded the evaluation (scale of 1–7, in which 1 is very, very bad and 7 is very, very good) [27].

### 2.5. Calculations of Training Indexes

Through stress, fatigue, DOMS and sleep quality, accumulated data by week were calculated, which includes the summation of each value provided by each training session. Through s-RPE, the following variables were calculated: (i) TM (mean of training load during the seven days of the week divided by the standard deviation of the training load of the seven days) [25,35];
$$\mathrm{TM}=\frac{\mathrm{mean}\;\mathrm{of}\;\mathrm{training}\;\mathrm{load}\;\mathrm{during}\;\mathrm{the}\;\mathrm{seven}\;\mathrm{days}\;\mathrm{of}\;\mathrm{the}\;\mathrm{week}\;}{\mathrm{standard}\;\mathrm{of}\;\mathrm{training}\;\mathrm{load}\;\mathrm{during}\;\mathrm{the}\;\mathrm{seven}\;\mathrm{days}\;\mathrm{of}\;\mathrm{the}\;\mathrm{week}};$$

(ii) TS (sum of the training loads for all training sessions during a week multiplied by training monotony) [25,35];
TS=sum of the training loads for all training sessions during a week∗TM;

Finally, (iii) ACWR (dividing the acute workload, i.e., the 1-week rolling workload data, by the chronic workload, i.e., the rolling 4-week average workload data) [38,39,40].
$$\mathrm{ACWR}=\frac{\mathrm{acute}\;\mathrm{workload}\;\left(\mathrm{most}\;\mathrm{recent}\;\mathrm{week}\right)\;}{\mathrm{chronic}\;\mathrm{workload}\;\left(\mathrm{last}\;4\;\mathrm{weeks}\right)}.$$

### 2.6. Statistical Analysis

Data were analyzed using SPSS version 22.0 (SPSS Inc., Chicago, IL, USA) for Windows. Initially, descriptive statistics were used to describe and characterize the sample. The Shapiro–Wilk and Levene tests were used to test the assumption of normality and homoscedasticity, respectively. Then, One Way ANOVA was used with the Bonferroni post hoc test to compare player positions and independent *t*-test was used to compare player status [41]. Pearson product–moment correlation coefficient was calculated between s-RPE and HI scores with the following thresholds: ≤0.1, trivial; >0.1–0.3, small; >0.3–0.5, moderate; >0.5–0.7, large; >0.7–0.9, very large; >0.9–1.0, almost perfect. Results were considered significant with *p* ≤ 0.05.

## 3. Results

The weekly changes in DOMS, stress, sleep and fatigue over the 10 week period are presented in Figure 1. Overall, DOMS presented the highest value in week 8 (13.6 arbitrary units (AU)) and the lowest in week 2 (6.8 AU); stress presented the highest value in week 6 (12.0 AU) and the lowest in week 10 (7.3 AU); sleep presented the highest value in week 8 (14.9 AU) and the lowest in week 2 (10.4 AU); and fatigue presented the highest value in week 8 (14.7 AU) and the lowest in week 2 (8.7 AU).

Training monotony and strain over the 10 week period are presented in Figure 2. Overall, training monotony presented the highest value in week 1 (6.3 AU) and the lowest in week 7 (2.7 AU); training strain presented the highest value in week 2 (9665.1 AU) and the lowest in week 7 (3957.6 AU).

ACWR over the 10 week period is presented in Figure 3. Overall, ACWR presented the highest value in week 5 (1.11 AU) and the lowest in week 10 (0.86 AU).

Table 2 presents comparisons for all measures between player positions, while Table 3 presents comparisons between player status. There were no significant differences between player positions nor status.

Considering the overall team, there were some associations between Hooper Index categories and s-RPE indexes. Specifically, the following correlations were found: stress and fatigue (0.693, *p* < 0.01); stress and DOMS (0.593, *p* < 0.01); stress and TS (−0.516, *p* < 0.05); fatigue and DOMS (0.688, *p* < 0.01). Table 4 presents all correlations.

## 4. Discussion

The aims of this study were: (a) to describe the weekly responses regarding internal training load parameters (TM, TS and ACWR) derived from perceived exertion; (b) to describe weekly variations of wellness markers [stress, fatigue, DOMS and sleep quality]; (c) to identify whether the (unknown) seasonal variations in both load and wellness measures are dependent upon playing position and status; and (d) to analyze possible associations between the several training measures and wellness markers. The main results of the study were: the peak on stress occurred in the week immediately subsequent to when the highest ACWR values were observed, while the lowest values of either stress or ACWR were reported concomitantly in the same week period; best self-reported sleep quality was identified simultaneously in the week where peaks of fatigue/DOMS occurred whilst the inverse also holds true, that is, the smallest indices of DOMS, fatigue and sleep quality temporally coincided; dynamics of perceived load and wellness variations during a 10-week in-season period were similar for starters and reserve players; and playing position was also not a factor affecting load distribution and wellness sensation in women’s soccer.

A slight spike in the ACWR of s-RPE (week 5) likely induced players into a greater perception of stress in the nearest next period (week 6). In the same response pattern, ACWR and stress indices reached the lowest magnitudes together in the end of the monitoring period (week 10). It is important to note that, despite ACWR derived from s-RPE not being a good predictor of injury [42], our outcomes observed across the in-season period did not reach the suggested “danger zone” (ACWR ≥ 1.5) [43]. This result is in line with another study performed with 65 players from Division I in the United States, which reported this ratio was not associated with injury [44], although some studies reported associations between ACWR and non-contact injury occurrences [42,45,46,47,48]. In fact, ACWR score has also been related to perceived effort in women athletes [49]. In addition, TMs across the monitoring weeks were all superior to the traditional cut-point of 2 AU [17]. Although different in magnitude, apparently there were four “valleys” (e.g., week 3 to 4 and week 9 to 10) and three peaks in TM/TS behaviors and this includes week 5 (Figure 2). In this way, women players may be sensitive to simultaneous changes (i.e., some increase) in ACWR, TS and TM—in particular when the first factor notably raises—thus reflecting an ensuing exacerbated stress sensation. As some studies have shown differences in the types of training depending on the moment of the season (i.e., pre, early, mid or end season) [25,50], it is possible link these to the body’s adaptations in attempting to accommodate a distinguished training intensity/duration (see Table 1) delivered in this mid-season moment. Owing to the existing relationship between stress symptoms and injury likelihood [51] as well as negative training-induced central adaptations [52], the close monitoring of women players is required at the time of which ACWR and stress spikes happen, aiming to avoid potential time-loss injuries and occasional performance declines.

The present work failed to identify significant relationships between players’ sleep quality, wellness perceptions and training indexes, although the dynamics of between-week changes in sleep accompanied the minimum and maximum values of both perceived fatigue and DOMS. Many scientific investigations have confirmed the direct impact of training intensity [53,54,55] and well-being [46,56] upon sleep measures in soccer. Here, the absence of meaningful correlations of sleep and all other dependent variables may be attributed to a potential lack of sensitivity provided by discrete, self-reported sleep quality metrics. As, for example, sleep questionnaire (subjective) responses do not often match actigraphy-derived or polysomnography (objective) parameters [57]. Nevertheless, when sudden changes are observed in both fatigue and DOMS (especially from week 7 to 8 where TS and TM also increased) attention is required because exacerbated levels of both fatigue and DOMS markers, as obtained by the Hooper Index method, may lead to impaired technical performance [58]. Interestingly, Douchet et al. [59] showed that a week with more accelerations and decelerations induced increased fatigue as observed in the present study by the greater RPE and Hooper index. The authors reported that the objectives of each training session during the week (i.e., technical, tactical, or physical) can contribute to defining the fatigue levels, this being an excellent way to manage the athletes’ training load [59].

On the other hand, this is the period in which women soccer players slept better, at least according to their reports. This could be associated with a necessity to restore the systemic homeostasis as illustrated by the elevated tiredness and pain sensations [60]. Therefore, according to the current results, encouraging sleep hygiene strategies could be of particular interest across an intensified training period. Otherwise, the women players choose to extend their sleep opportunities to improve the recovery process and prevent occasions of poor performance in training sessions and official matches due to the high levels of fatigue and muscle soreness. Of note, when the training duration was shortened (e.g., weeks 2 and 7), TM and TS were also impacted in the same direction of declines, thus suggesting that it could be a reasonable strategy when it is necessary to adequately recover players. Notwithstanding, TM and TS peaked respectively in weeks 1 and 2, which is similar to research into male soccer [24,36] and might illustrate the challenge of female athletes to deal with training demands imposed at the beginning of a season.

A key finding of the present investigation was that variations of selected internal load parameters and wellness responses were independent of playing status. Positional role also had a minimal influence on the in-season fluctuations discussed above. One possible explanation for these results could be associated with very small subsample sizes of four to five players, which was not enough to demonstrate significant differences, as demonstrated by previous studies [25,61]. Locomotor capacity and the recovery states of women soccer players are sensitive to the accumulated load demands [62] implying that players’ rotation might be needed to counteract such worst-case conditions (e.g., when training was supposedly intensified and/or extended; weeks 5 and 8). In this sense and despite there being players who were selected as starters more often, there should be changes in the coaches’ choices depending on these intensified periods. This approach will help to reduce the differences between the playing status in the squad. Therefore, and despite there being players who are the preferred starters, regular changes in the main squad may occur as a function of such intensified periods, which helps to justify the lack of differences according to statuses. While the absence of differences in the analysis of starters versus reserve players is in agreement with the current evidence [24,50], the playing position had a significant influence in an almost matched male sample considering Hooper’s Index measures [30]. In this sense, it is necessary to point out the possibility that, in women’s soccer, the traditional position-related outputs might be more limited or even non-existent. To be explicit, fitness testing data of past studies indicated that aerobic power was similar across outfield playing positions in Norwegian elite women players [63]. Furthermore, research on college players from Division I in the United States demonstrated that lower limb power, change-of-direction [64], agility, speed and acceleration qualities [65] did not differ according to positional role. Recently, it was confirmed that some of these tests may reflect the in-game running outputs in first Division league Portuguese women soccer players [31], thus suggesting that a systematic discrepancy in game demands depending on player position would not be so evident in females, which could be observed in the study of Vescovi et al. [32], as compared to substantial between-position differences generally found in male counterparts [66,67]. More specific to the present context, external training load in a first Division Spanish women’s team revealed no significant differences as a function of playing position (central defenders, wide defenders, central midfielders, wide midfielders and strikers) [68]. Results from a meta-analysis confirm the internal–external training load associations in team sports [10], thereby again making our result concerning training outcomes being independent of playing position compatible with the literature. In sum, it seems that modern soccer demands cause adjustments by the coaching staff in order to deliver relatively equalized training stimuli to women players regardless of their positional role and whether beginning the matches in the starting line-up or on the bench.

Finally, an inverse relationship existed between stress and RPE in the monitored population of women elite soccer players. It was the only significant correlation identified between s-RPE and Hooper Index categories, which contradicts two previous studies in male professional soccer players reporting a number of direct associations amongst RPE and Hooper Index indices [33,69]. However, the large negative s-RPE-stress correlation found here is something that provides preliminary evidence of the possible harmful consequences of higher stress levels on the perception of degree of efforts expended in practice sessions. This inverse relationship seems to be in line with a recent study that reported higher values of RPE and Hooper index categories with the exception of stress during a higher intensity week [59]. This is in accordance with the notion of Paul et al. [70], where it cannot be discarded that high stress levels may promote a suboptimal psychological interaction with the question being asked relating to perceived effort. As a result, when collecting s-RPE in the context of women’s soccer, it is important to consider the prominent interference of stress values that could potentially underestimate athletes’ perceived training load demands.

Our present study provides coaches and technical staff with recent knowledge about the weekly variations of TM, TS and ACWR through s-RPE, and the Hooper index categories, in order to have all players available for competition. In addition, this study allows coaches to understand that all these measures vary according to the intensity and duration of the sessions throughout the week. Therefore, planning the structure and periodization of the objectives (i.e., technical, tactical, or physical) and the use of measures of intensity (e.g., RPE) are essential to inducing good adaptations in female athletes.

Limitations of the current study should not be overlooked and includes the fact that: (i) data collections encompassed a single-club, suggesting caution in attempts of extrapolating results to a variety of other teams/leagues; (ii) there was a relatively small sample size when the women athletes were grouped into distinct outfield playing positions; (iii) fatigue levels were measured only via self-reported ratings but not gold standard measures; (iv) non-concomitant consideration for situational factors that can modulate internal load responses during a season; and (v) a lack of match load information when interpreting training outcomes.

## 5. Conclusions

To conclude, the in-season spike in ACWR calculated through s-RPE may induce women soccer players into experiencing a nearby subsequent peak of perceived stress. Additionally, higher levels of fatigue occur with levels of stress and higher levels of DOMS occur with high levels of stress and fatigue. Reducing training duration could diminish s-RPE-derived strain and monotony indices. Importantly, any in-season variations across a 10-week period concerning internal load and perceived wellness seems independent of position and status in outfield players, although some caution should be taken into consideration. Finally, the higher the players’ reported stress, the lower the observed s-RPE, thus possibly indicating a mutual interference of experienced stress levels on the assimilation of training loads by women professional soccer players.

## Figures and Tables

**Figure 1 ijerph-18-12817-f001:**
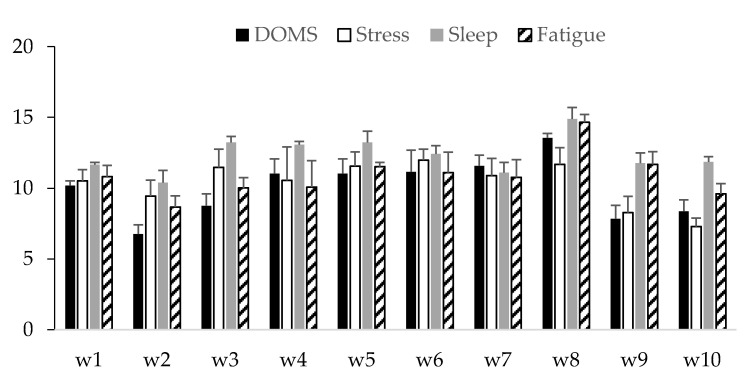
Description of weekly DOMS, stress, sleep and fatigue during the 10 weeks in AU (arbitrary units).

**Figure 2 ijerph-18-12817-f002:**
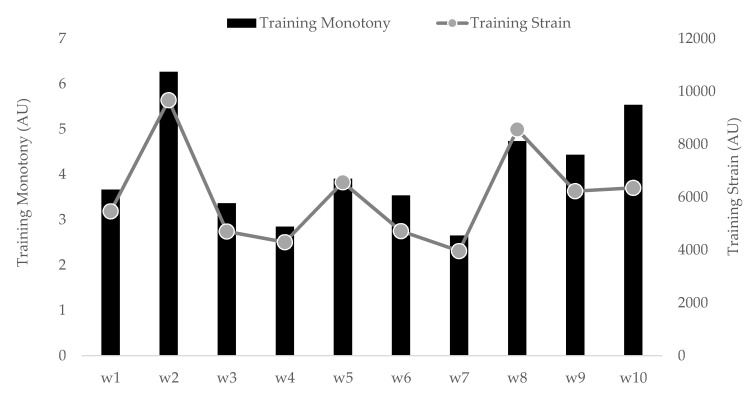
Description of training monotony and training strain during the 10 weeks.

**Figure 3 ijerph-18-12817-f003:**
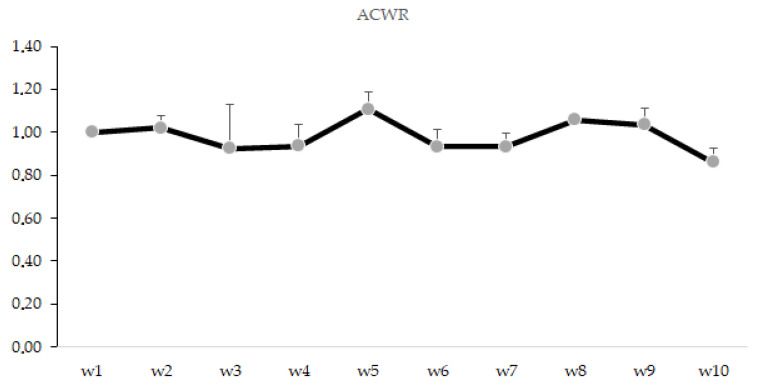
Description of ACWR during the 10 weeks in AU (arbitrary units).

**Table 1 ijerph-18-12817-t001:** Training sessions during the 10-week period.

Weeks (w)	w1	w2	w3	w4	w5	w6	w7	w8	w9	w10
Session duration (total minutes)	385	250	294	285	317	280	274	316	331	270

**Table 2 ijerph-18-12817-t002:** Descriptive statistics (mean ± SD) of weekly stress, fatigue, DOMS, sleep, training monotony, training strain and ACWR between playing positions.

Measures (AU)	Defenders	Central Midfielders	Wide Midfielders	Strikers	F	*P*
Stress	12.1 ± 2.8	8.9 ± 1.1	10.7 ± 1.2	9.8 ± 2.8	1.830	0.185
Fatigue	11.5 ± 2.2	9.9 ± 0.6	11.4 ± 1.8	10.7 ± 2.4	0.744	0.542
DOMS	9.7 ± 2.2	9.2 ± 2.2	11.2 ± 2.4	9.7 ± 2.7	0.513	0.680
Sleep	12.9 ± 1.5	12.6 ± 0.9	11.9 ± 1.1	12.2 ± 1.3	0.589	0.632
TM	4.3 ± 0.6	4.6 ± 1.0	4.9 ± 0.3	4.4 ± 0.9	0.394	0.759
TS	82.3 ± 9.3	176.4 ± 167.2	125.9 ± 66.4	199.2 ± 258.3	0.518	0.676
ACWR	0.97 ± 0.02	0.97 ± 0.03	0.96 ± 0.01	0.98 ± 0.03	0.240	0.867

Abbreviations: AU, arbitrary units; DOMS, delayed onset muscle soreness; TM, training monotony; TS, training strain; ACWR, acute: chronic workload ratio.

**Table 3 ijerph-18-12817-t003:** Descriptive statistics (mean ± SD) of weekly stress, fatigue, DOMS, sleep, training monotony, training strain and ACWR between playing status.

Measures (AU)	Starters	Non-Starters	T	*P*
Stress	10.7 ± 2.8	9.8 ± 1.6	0.808	0.430
Fatigue	10.9 ± 2.0	10.8 ± 1.9	0.157	0.877
DOMS	10.5 ± 2.7	9.1 ± 1.2	1.339	0.198
Sleep	12.5 ± 1.4	12.2 ± 0.9	0.482	0.636
TM	4.3 ± 0.7	4.8 ± 0.8	−1.404	0.178
TS	137.5 ± 173.9	160.1 ± 135.2	−0.306	0.763
ACWR	0.96 ± 0.02	0.98 ± 0.03	−1.263	0.224

Abbreviations: AU, arbitrary units; DOMS, delayed onset muscle soreness; TM, training monotony; TS, training strain; ACWR, acute: chronic workload ratio.

**Table 4 ijerph-18-12817-t004:** Correlation analysis between measures for overall team.

Measures	β0	β1	β2	β3	β4	β5	β6
Stress (β0)	1.00						
Fatigue (β1)	**0.693**	1.00					
DOMS (β2)	**0.593**	**0.688**	1.00				
Sleep (β3)	0.412	−0.037	−0.001	1.00			
TM (β4)	−0.155	0.165	−0.996	−0.248	1.00		
TS (β5)	**−0.516**	−0.411	−0.329	−0.365	0.043	1.00	
ACWR (β6)	−0.071	0.133	0.148	−0.057	0.012	0.300	1.00

Correlations are highlighted in bold (*p* ≤ 0.05). Abbreviations: DOMS, delayed onset muscle soreness; TM, training monotony; TS, training strain; ACWR, acute: chronic workload ratio.

## Data Availability

The data presented in this study are available on request from the corresponding author.

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
