# Peer review of "In-Season Internal Load and Wellness Variations in Professional Women Soccer Players: Comparisons between Playing Positions and Status"

_ijerph, 2021, doi:10.3390/ijerph182312817_

Round 1

Reviewer 1 Report

Dear authors, I attach my considerations:

Abstract

The abstract adequately summarises the research, including the objective, the methodology used, the main findings of the results and the most relevant conclusions.
However, in the use of acronyms, each letter of each word should be capitalised as follows:
rated perceived exertion (s-RPE) Rated Perceived Exertion (s-RPE);

Keywords:
I recommend that the keywords are changed to words that do not appear in the title of the paper. This will multiply the search options. Also, do not include acronyms that have not been explained above. 

Introduction

The general idea of the research is presented with an adequate and updated theoretical framework.

Correct all acronyms and abbreviations as indicated in the abstract: rated perceived exertion (RPE) - line 80-.

Introduce examples from other women's team sports research to determine the progress of this technology globally (e.g. hockey, rugby, basketball, handball and beach handball, etc.).

2. Materials and Methods
2.1. Subjects

Justify with other similar studies and research methodology studies the number of subjects selected. There are 19, why?

Results 

They are clear.

Discussion 

Lines 218-237 are not discussed with any other studies. The study approach, results and objectives are repeated. This is not necessary. Even conclusions are drawn, this is not the section.

In the discussion all statements should have previous research that supports or contradicts the results obtained. This issue should be reviewed.

Conclusions
They are clear.

References
Are adequate

Reviewer 2 Report

Dear authors,

the manuscript was well written, to my opinion. Nevertheless, I found some typographic or grammar mistakes, I'm affraid.

I have made colored marks in the PDF for those minor flaws. Please see the attachment for that.

I also made some notes in the PDF where I have to put my finger on some problems that might have a greater impact for ciriticisms.

Please, let me give some examples here.

A smaller point is that you use your abbreviated variable at the end of the introduction (l. 119) without giving the explanation - this was given much later in the discussion (l. 21-220). Please find a suitable sulution.

The variables TM, TS, ACWR are not quite simple to understand. Probably you find a way for an illustration for a better comprehensive understanding.

Watching your figures, I wonder wether or not there should be error bars in fig. 2+3? If I understood right, the presented variables contained s-RPE values of the sample (n=19); and in the following tables descriptives are given as mean and standard deviation. Please clarify.

Referring also to the results section, I think it should be worth it to report not only the descriptive statistics in table 2+3, but also the respective F- or t- and p-values, because you reffer to these non-significant results in your discussion (l. 284 + 286).

The same problem occures when you mention non-significant correlation analyses results (e.g. sleep with something) in the discussion (l. 258) without giving the respective correlation coefficients in the results section (l. 213-216). I would like to recommend an additional table (matrix) for all the correlation analyses enabeling the reader to follow your discusson and conclusion.

I apreciate very much that you reported the limitations of your investigation, but I think you should find a way to make a note that your "key finding" (l. 284) was at least partly due to your very small subsamples of 4-5 individuals when you run your 1-way ANOVA in oder to identify differences between player positions. That should not be overlooked before you discuss this lacking significance in face of the current literature (from a statistical point of view, one should be extremely careful with the discussion of non-significant findings).

Please consider my recommendations or give a short rebuttal.

Thank you.

Reviewer 3 Report

Reviewer comments.

The current paper evaluates wellness indicators for professional women soccer players.  As a soccer fan, it’s exciting and refreshing to see research on women athletes.  However, there are some limitations in the current manuscript.

In general, some of the grammar and syntax are incorrect, detracting from evaluating the scientific aspect of the paper.  I have highlighted some, but a full review with this in mind is recommended.

Impact.  What is the learning of these data? It wasn’t clear to me to whom the authors would communicate their results.  To coaches, to change their training?  To players?  Others?  Further, what is the objective to conduct this research.  It is to reduce injury or increase performance?  Is there some wellness, satisfaction component.  I suggest the authors add some text to offer the impact/value of there research.

Introduction. 

Lines 47 – 54.  I don’t understand this paragraph.  What is load monitoring?  Reword “providing some light”

Lines 62 – 90.  The difference between external load and internal load is lost on me.  In this study, the authors are only using self reported outcomes.  This section should be rewritten more clearly and justifying the approach taken in the current study.

Lines 92.  Reword “such a fact”….if it has been dismissed, it’s not a fact.

Line 99.  Reword “such a fact”

Lines 102 – 105.  Reword this sentence.  I don’t think “once wellness is multiple dependent” is correct.  What is expectable”

Lines 114  - 116.  Reword this sentence.  It is confusing and uses incorrect grammar.

Line 142.  Omit “before Christmas”.  Why is this important?

Table 1.  This is confusing.  Are the minutes total for the week or average per session.  Do the players have 6 hour sessions per week or 3 two-hour sessions?  Please clarify in the title and text.

Wellness quantification.  The options are 1 – 7 for each.  (lines 160 – 165).  How does this translate to the range of 0 – 20 in Figure 1?  Please clarify in the title and text.  Further, Figure 1 is difficult to interpret because DOM, stress and fatigue relate to poor outcomes with high numbers but the opposite is true for sleep.  Suggest making the direction the same for all outcomes.

Lines 252.  What is distinguished training?

Lines 258 – 260.  Reword this sentence.  It is confusing and uses incorrect grammar.

Lines 274 – 277. Reword this sentence.  It is confusing and uses incorrect grammar.

290 – 293.  Reword this sentence.  It is confusing and uses incorrect grammar.

  1. The “research in Division 1 college (e.g. NCAA)” is not right. It would be better to say,” research of college players in the US…”.  The NCAA covers college play at all levels, the Divisions refer to scholarships.

Line 301.  Change the verb to “differ”

Line 304.  Reword “which you can see”

Round 2

Reviewer 2 Report

Dear authors,

I hope my recommendations and suggestions were helpful to improve the manuscript.

I appreciate very much your modifications and I accept your rebuttal concerning the error bars of some of your figures.

Congratualtions, from my point of view I am fine with the manuscript in its preset form.

Best wishes,

Jan

Reviewer 3 Report

The manuscript is much improved following the recommended revisions.  A few minor suggestions.

Table 1 is still confusing.  Suggest changing the row title to "accumulated training duration" since session duration is incorrect.

Table 2.  P should not be capitalized.  Suggest title the column as "p-value".

Line 386, note that Division 1 in the US are college players.